# Closed-Loop Recycling of Poly(vinyl butyral) Interlayer Film via Restabilization Technology

**DOI:** 10.3390/polym17030317

**Published:** 2025-01-24

**Authors:** Vasilis Nikitakos, Athanasios D. Porfyris, Konstantinos Beltsios, Rudolf Pfaendner, Beatriz Yecora, Angelica Perez, Filip Brkić, Filip Miketa, Constantine D. Papaspyrides

**Affiliations:** 1Laboratory of Polymer Technology, School of Chemical Engineering, Zographou Campus, National Technical University of Athens, 15780 Athens, Greece; vnikitakos@mail.ntua.gr; 2Laboratory of Materials Science & Engineering, School of Chemical Engineering, Zographou Campus, National Technical University of Athens, 15780 Athens, Greece; kgbelt@mail.ntua.gr; 3Fraunhofer Institute for Structural Durability and System Reliability LBF, Division Plastics, 64289 Darmstadt, Germany; rudolf.pfaendner@lbf.fraunhofer.de; 4LUREDERRA Technological Centre, Perguita Industrial Area, 31210 Los Arcos, Spain; beatriz.yecora@lurederra.es (B.Y.); angelica.perez@lurederra.es (A.P.); 5Bio-mi Društvo s Ograničenom Odgovornošću za Proizvodnju, Istraživanje i Razvoj, Put Brdo, 51211 Matulji, Croatia; filip.brkic@bio-mi.eu (F.B.); filip.miketa@bio-mi.eu (F.M.)

**Keywords:** polyvinyl butyral, restabilization, recycling

## Abstract

Polyvinyl butyral (PVB) is a specialty polymer primarily used as an interlayer in laminated glass applications with no current circularity plan after the end of its life. This study presents a comprehensive recycling strategy for postconsumed PVB wastes based on a remelting–restabilization approach. Thermo-oxidative degradation of PVB was analyzed under heat and shear stress conditions in an internal mixer apparatus. The degradation mechanism of plasticized PVB (p-PVB) and unplasticized PVB (u-PVB) was identified as chain scission through melt flow rate (MFR), intrinsic viscosity (IV), and yellowness index (YI) characterization. Six different antioxidant (AO) formulations were screened for their effectiveness in inhibiting degradation in both neat u-PVB and p-PVB, as well as retrieved PVB. The phenolic antioxidants 1,3,5-trimethyl-2,4,6-tris(3,5-di-tert-butyl-4-hydroxybenzyl)benzene and 4-[[4,6-bis(octylsulfanyl)-1,3,5-triazin-2-yl]amino]-2,6-di-tert-butylphenol were found to be the most effective ones based on MFR, oxidation onset temperature (OOT), and YI evaluations, while the optimal AO concentration was determined at 0.3% *w*/*w*. Furthermore, upscaling of the process was achieved by mixing virgin PVB and high-quality retrieved PVB wastes with AOs in a twin-screw extruder. Testing of the recycled samples confirmed that the selected AOs offered resilience against degradation at reprocessing and protection during the next service life of the material.

## 1. Introduction

Polyvinyl butyral (PVB) is an amorphous thermoplastic terpolymer dominating the sector of laminated glass applications [1]. The three repeating units of the polymer are vinyl butyral (VB) at ca. 70–85%, vinyl alcohol (VA) at ca. 10–25%, and vinyl acetate (VAc) at ca. 1–5% [2,3,4]. Its chemical structure enables strong adhesion to glass via the hydroxyl groups of the VA segment, which adhere to the glass silanols, thus providing the required protection to the users by retaining glass fragments after a collision [5,6]. Pristine PVB is a rigid polymer with a glass transition temperature (*T*_g_) in the range of 70–80 °C [7,8]. Commercially, it is mostly found in the form of plasticized interlayer films, usually plasticized with triethylene glycol bis(2-ethylhexanoate) (3GO) or dibutyl sebacate (DBS) in the range of 20–40% *w*/*w* [3]. This plasticization leads to a *T*_g_ reduction of ca. 50–70 °C, thus resulting in improved flexibility, processability, and impact resistance [9,10]. Plasticized PVB is flexible, transparent, able to attach to silicate glasses, and stable to long-term sunlight exposure (at least when covered by glass), thus being appropriate for the role of a polymeric interlayer in laminated glass used for automotive or construction applications, i.e., for windscreens and safety windows, respectively [6]. The development of polyvinyl butyral (PVB) has been primarily driven by the need for safety in the automotive and architectural industry, contributing in 2017 to shares of 52.02% and 42.83%, respectively [11]. PVB resins are also being consumed in other minor applications: coatings, wash primers, textiles, structural adhesives, and solid oxide fuel cells [12,13]. The global PVB market is valued now at approximately 9.13 billion EUR, and it is expected to reach around 13.07 billion EUR by 2031 with a compound annual growth rate (CAGR) of about 4.60% from 2024 to 2031 [11]. While PVB use is on the rise, there is no active comprehensive recycling strategy so far for it. Glass recyclers across Europe focus solely on recovering the residual glass from laminated glass wastes, neglecting the PVB component. PVB material in laminated glass is incinerated or landfilled, and only 9% is recycled, mostly for secondary uses. Every year in Europe, about 480,000 tons of laminated glass wastes result from end-of-life vehicles (ELVs) [14]. Up to now, to the best of our knowledge, there is no holistic approach aiming at the closed-loop recycling of PVB, while it remains a valuable polymer (€5.24/kg), and its reclamation within the circular economy framework could be ecological and lucrative [14].

The complexity and cost of recycling PVB wastes create a variety of problems that may have an impact on market demand and the overall accomplishment of this aim. Extracting PVB from laminated glass, such as windscreens and architectural glass, might be a time-consuming and difficult operation including shredding the glass and PVB mixture followed by mechanical and chemical separation processes to separate the PVB and purify the recycled PVB to fulfill quality criteria for reuse. In the previous literature, Swain et al. utilized a recycling process for laminated glass wastes by shredding and milling to extract glass. To completely remove glass residues, they applied a nonionic surfactant and studied the separation time and stirring speed of the process [15]. In other studies, Tupy et al. suggested a three-stage technology for recycling PVB, resulting in sheets with minimal residual glass content (up to 300 ppm) and minimal change in properties. However, it was found that while increasing alkali ion concentration in the delamination process could loosen the adhesion with glass, it can also cause the saponification of the polymer’s plasticizer, which is considered undesirable in the recycling process [16]. The research group of Soos et al. also studied the criteria for an effective high-efficiency separation technology from multilayer glued glasses, conducting breakage and vibration tests and washing the interlayer film to minimize noise and dust of the process, ensuring greater cleanliness of the individual type of waste [17,18]. A different approach was adopted by Guner et al. using the dissolution precipitation method as an approach to PVB recycling. A portion of the waste windscreen was dissolved in tetrahydrofuran (THF), and then PVB was precipitated with n-hexane, achieving an overall low separation efficiency of up to 8% [19]. In addition to this, recycling of PVB has been noted on an industrial scale by producing dispersions used in secondary applications such as carpets, paints, textiles, and building materials [20]. An innovative and efficient mechanochemical treatment of laminated glass is herein implemented to obtain highly pure retrieved polyvinyl butyral (PVB) with minimal residual glass (<0.5% *w*/*w*), which is considered negligible for industrial PVB sheet reprocessing [21].

Nevertheless, apart from the delamination challenges concerning the glass residues in PVB wastes, which might cause haze and turbidity, there are other issues associated with the large-scale and closed-loop recycling of PVB. Taking PVB recycling on a large scale will result in variability of PVB interlayer compositions, different plasticizer types and contents, different interlayer functions (acoustic or solar), and different levels of degradation. These variations might impact the transparency, mechanical, or adhesion performance. For this reason, within a previous study, an integrated characterization system of PVB was introduced [3]. The use of an optical multisensor tool to separate and categorize the different grades of laminated glass has rendered the management of PVB wastes successful and efficient [3]. Furthermore, laminated glass after its end of life (EoL) has been exposed to outdoor conditions and ageing, while its recycling might be delayed. For that reason, it is crucial to assess the degradation state of PVB interlayers that have been exposed to outdoor conditions (UV, heat, humidity).

Additional heat degradation of PVB due to elevated temperatures and shearing during melt reprocessing is also expected. In the process of recycling, these conditions can induce yellowness and deterioration of molecular weight, and consequently, of mechanical properties. Studying the degradation of PVB is a prerequisite stage for a successful high-end product in the recycling process aiming at closed-loop recycling. Several studies have been conducted on the degradation of PVB, proposing that it is subjected to scission of the macromolecules during shear and thermal degradation [7,8,22,23,24,25,26,27,28,29,30,31]. The suggested mechanism of thermal degradation of PVB pinpoints the initial point of degradation to be the acetate group, which is the weakest center. This was experimentally proved by El-Din et al. by comparing the acetate content of an unheated sample (blank sample) with the acetate contents of samples heated to various temperatures (50–200 °C) and indicating an increase in the characteristic peak of carbonyl via FTIR at ca. 1730 cm^−1^ [22]. Furthermore, Liau et al. reported three stages of degradation for PVB in a nitrogen atmosphere, but four stages were observed in the air. A possible free-radical mechanism of PVB thermal oxidative degradation was proposed, with major products of the thermal oxidative degradation being PVB; CO, butanal, butenal, acetic acid, butanoic acid, and butanol. In addition to this, usually, some acids, such as hydrochloric acid and butanoic acid, would be formed during the synthesis of PVB, and these acid substances are inclined to cause the opening of the 1,3-dioxane ring of PVB at elevated temperatures [28].

Polymer degradation occurs not only at the reprocessing stage but during the entire life cycle of the polymer product through an auto-oxidative free radical chain reaction process [32,33]. The reaction of organic compounds with molecular oxygen is called autoxidation because such reactions can proceed automatically whenever these compounds are exposed to the atmosphere [33]. The process begins with a reactive site, a free radical formed on the polymer chain reacting to oxygen to form a peroxy radical (ROO*). This peroxy radical further abstracts a hydrogen atom from the polymer chain, creating hydroperoxides (ROOH) and generating new free radicals in a chain reaction [32]. The cycle continues as these radicals propagate, leading to further breakdown of the polymer and eventually leading to the deterioration of the polymer’s mechanical (embrittlement) and optical properties (color, haziness). Autoxidation can be inhibited by commercial additives, i.e., antioxidants (AOs). AOs are added at the compounding step, right before or during the polymer processing phase, and they are consumed partly at this stage but partly during the service life of the product to protect it from degradation and therefore enhance its longevity [34,35,36]. Primary AOs (e.g., hindered phenols) quench peroxy radicals (ROO*) by donating their protons, while secondary AOs (e.g., phosphites or thioester moieties) decompose the produced hydroperoxides and overall inhibit the oxidative process [37]. Remelting restabilization technology is an essential process in thermoplastic polymer recycling, in which readditivation of the recycled polymer stream with appropriate AOs results in maintaining the material’s quality and extending its useful life for a wide range of applications [35]. As indicated by previous work by Kartalis et al., via this technology, it is feasible to restabilize the polymer, maintaining its current state and inhibiting any further degradation suffered during reprocessing and/or the second life cycle of the material [35,38,39]. HDPE, PE, and PP have been extensively studied for their mechanical recycling, degradation mechanisms, and restabilization [40,41,42,43,44,45,46,47,48].

In this paper, the degradation mechanism of PVB was studied by subjecting the polymer to thermo-oxidative degradation (heat and shear stress) under experimental conditions that mimic real industrial reprocessing conditions. The resulting degradation was mitigated by the addition of appropriate combinations of primary and/or secondary AOs. For this purpose, various PVB-AO formulations were tested, with loading concentrations ranging from 0.1% to 0.5% by weight, using melt blending in an internal mixer apparatus. Different residence times were examined, and AO performance against thermo-oxidative degradation was evaluated via the determination of melt flow rate (MFR), dilute solution viscometry (DSV), yellowness index (YI), and oxidation onset temperature (OOT). Based on the combination of these analytical techniques, the most appropriate AO formulation to be exploited in the closed-loop recycling process of PVB wastes was determined.

## 2. Materials and Methods

A commercial grade (Saflex RB41) of plasticized PVB (p-PVB) containing triethylene glycol bis(2-ethylhexanoate) as plasticizer was supplied by Eastman Chemical Company (Kingsport, TN, USA). Unplasticized PVB (u-PVB) was produced by American Polymer Standards Corporation, Mentor, OH, USA and supplied by Eastman Chemical Company. High-quality retrieved PVB grades, i.e., re-PVB, with a very low residual glass content (<0.5 wt%) were delaminated from various postconsumer laminated glass wastes through a patented mechanochemical process [21].

For the remelting–restabilization technology, the incorporated antioxidants (AOs) for inhibition of thermo-oxidative degradation of PVB are presented in Table 1 and Appendix A. All antioxidants were kindly supplied by BASF Hellas S.A (Athens, Greece).

PVB is mildly hygroscopic, and water can act as a plasticizer, increasing slightly the value of MFR; therefore, a drying protocol at 50 °C for 6 h under a static vacuum of ca. 600–800 mbar was applied to all samples to remove excess moisture [24,49].

A Banbury-type internal mixer T300B (Brabender, Duisburg, Germany) was used for the incorporation of the antioxidants in u-PVB, p-PVB, and re-PVB samples, as well as for the degradation experiments. Dried batches of 50 g of each PVB grade were loaded in the preheated (150 °C) chamber and were processed for 5, 10, and 15 min at a constant screw speed of 40 rpm. Subsequently, the mixer was discharged, and the PVB material was cooled down, ground down to pellets, and further dried in vacuo before any further characterization.

### 2.1. Fourier Transform–Infrared Spectroscopy (FT-IR)

ATR (Attenuated Total Reflection) FT-IR spectroscopy measurements were performed on a Bruker AII ATR Spectrometer (Bruker Corporation, Billerica, MA, USA) from 400 to 4000 cm^−1^ with a 4 cm^−1^ resolution using a diamond crystal. Thirty-two coadded spectra were taken for each measurement. All samples characterized via FT-IR were in the form of films of ca. 0.70 mm thickness.

### 2.2. Thermogravimetric Analysis (TGA)

The thermal stability of the PVB samples was determined by thermogravimetric analysis (TGA) in a Mettler Toledo TGA/DSC HT 1 (Mettler-Toledo International Inc., Greifensee, Switzerland) apparatus. The samples were analyzed through dynamic heating from 30 to 600 °C under controlled nitrogen flow (10 mL/min) at a heating rate of 10 °C/min. The onset of decomposition was defined as the temperature at 5% weight loss (*T*_5%_), the degradation temperature (*T*_d_) was determined at the maximum rate of weight loss (1st derivative curve), and the char yield was determined as the *w*/*w* % residue left at 600 °C. Measurements were performed in triplicate for all received PVB samples, and the average value corrected by the standard deviation of the measurements was extracted.

### 2.3. Differential Scanning Calorimetry (DSC)

Initial materials were characterized via DSC, applying heating–cooling–heating cycles from −40 to 150 °C at a heating (cooling) rate of 10 °C/min. The measurements were performed in a Mettler DSC 1 STARe (Mettler-Toledo International Inc., Greifensee, Switzerland) system under controlled nitrogen flow at 20 mL/min. The glass transition temperature (*T*_g_) was determined from the 2nd heating curves. In addition, the oxidation onset temperature (OOT) of all PVB compounds (with and without AO) was determined by separate DSC measurements from 30 to 300 °C at a heating rate of 10 °C/min under air atmosphere controlled at 50 mL/min [50,51,52].

### 2.4. Melt Flow Rate (MFR)

Melt flow rate (MFR) measurements were conducted for all unstabilized and restabilized PVB samples prepared according to ASTM D1238-10 [53]. The measurements were performed in a Kayeness Dynisco 4004 (Dynisco Europe GmbH, Heilbronn, Germany) rheometer at 190 °C, with a load of 10 kg for the u-PVB and 2.16 kg for the p-PVB. All the samples were dried prior to each measurement to exclude plasticization from the water [49].

### 2.5. Colorimetry

The tristimulus values of color space (X_CIE_, Y_CIE_, and Z_CIE_) were recorded using a calibrated handheld Hach Lange LMG 183 spectrophotometer (Konica Minolta, Tokyo, Japan). To accurately quantify yellowness, the YI was calculated according to Equation (1) [54].(1)$$YI=\frac{1.3013\times X_{CIE}-1.1497\times Z_{CIE}}{Y_{CIE}}\times 100$$

Films for YI evaluation were prepared by compression molding (at 140 °C and 50 bar for 5 min). The films were prepared to have similar thickness (ca. 0.70 mm) to accurately calculate YI and compare between different samples. YI was calculated at least in 10 different points of each film, and the average measurement was used. Furthermore, ΔYΙ (Equation (2)) was used to monitor the evolution of YI in the reprocessed compounds (5, 10, 15 min residence time) for the determination of optimal AO concentration.ΔYI = YI_1_ − YI_0_(2)

YI_0_ represents the average value of the yellowness index before the degradation test, while YI_1_ represents the average value of the yellowness index after the degradation test. A positive (+) ΔYI indicates increased yellowness, and a negative (−) ΔYI indicates decreased yellowness or increased blueness.

### 2.6. Dilute Solution Viscometry (DSV)

Solution viscometry was used to define the intrinsic viscosity (IV, [*η*]) of the PVB samples. Dilute solutions of 0.5 g·dL^−1^ concentration of each PVB sample in tetrahydrofuran (THF) were prepared. Especially for the plasticized grades, p-PVB and re-PVB, the concentration was corrected according to the plasticizer content as determined by TGA analysis. Measurements were performed in an Ubbelohde-type viscometer (K = 0.009340 mm^2^·s^−2^) at 30 ± 0.1 °C [1]. The outflow times of THF and each PVB solution were measured, and intrinsic viscosity [*η*] values were obtained by single point measurement via Equation (3) [55].(3)$$\left[\eta \right]=\frac{\sqrt{1+1.5\eta_{sp}}+1}{0.75C}$$

### 2.7. Pilot Production of Re-PVB Film

Flakes of re-PVB and virgin commercial plasticized PVB grade (Trosifol Clear, Kuraray) were first shredded in the HSS400-A granulator (Ningbo Chunnoo Machinery Co., Ltd., Ningbo, China) and then dried in a hot air drier at 60 °C for 6 h. After drying, they were shredded once again and compounded with antioxidants on the TSE35B twin screw extruder (Nanjing Haisi Extrusion Equipment Co., Ltd., Nanjing, China) with a screw diameter of 36 mm and an L/D ratio of 44. All compounds were melt-mixed on a constant temperature profile of 180 °C, a screw speed of 300 RPM, and a stainless-steel filter screen with 24 × 110 mesh and a hole diameter of 100 µm. Approximate residence time was measured with a colorant test to be around 3 min. The obtained granules were once again dried in a hot air drier at 60 °C for 6 h. The granules were extruded on an XH-432-30 cast extruder (Guangdong Xihua Machinery Co., Ltd., Dongguan, China) equipped with a 400 mm flat film die and an L/D ratio of 33. All films were extruded with a rising temperature profile from 120 °C up to 130 °C, with a screw speed of 90 RPM, and drawn with rolls cooled at 13 °C. Additionally, a low-density polyethylene film was used as an intermediate layer to prevent film blocking. The approximate residence time of the material was determined to be around 1 min. The received films were tested on the DZ-101 universal testing machine (Dongguan Zonhow Test Equipment Co., Ltd., Dongguan, China) for their mechanical properties following the ASTM D882 procedure [56]. The film samples were cut into rectangular stripes of 250 × 15 mm^2^ dimensions; the initial distance between the grips was 50 mm, and a grip separation rate of 500 mm/min was used to obtain tensile strength at break and elongation at break. Nine samples of each grade were tested, and the results were represented as their average. All samples were preconditioned at 22 °C and 50% RH for 40 h prior to measurement.

## 3. Results and Discussion

### 3.1. Characterization of Reference PVB Materials

FT-IR analysis was initially used to determine the critical characteristic peaks of PVB. Accordingly, as shown in Figure 1, all PVB samples exhibited two discrete peaks at about 1140 and 996 cm^−1^ that corresponded to C-O-C stretching vibrations of butyraldehyde groups in the vinyl butyral segment [2]. A broad peak related to hydroxyl content (3490 cm^−1^) was also present in the spectra of all PVB samples (u-PVB, p-PVB, and re-PVB), referring to vinyl alcohol (VA) content [4]. As for the spectra of the pure plasticizers 3GO (triethylene glycol bis(2-ethylhexanoate) and DBS (dibutyl sebacate), they exhibited different peaks than PVB in the range of 1000–1500 cm^−1^, which are attributed to their different chemical structures, but they cannot be easily determined when mixed with PVB because of the overlapping of the characteristic vibrational modes [3]. The most characteristic peak is found at 1740 cm^−1^ for both plasticizers, representing the C=O stretching, which is indicative of an ester moiety [3]. Regarding the FT-IR analysis of the plasticized PVB grades (Figure 1), the peak assignment involved characteristic peaks of the PVB terpolymer (VA, VB, VAc segments), as well as one at ca. 1740 cm^−1^ proving the presence of the plasticizer. Additionally, a low intensity for the referred peak was observed in the spectrum of the unplasticized sample (u-PVB) corresponding to the low vinyl acetate (VAc) content (1–5%) of PVB. Regarding the two plasticized samples, p-PVB and re-PVB presented no differences in their FTIR spectra, thus suggesting that PVB wastes shared the same characteristic peaks as commercial grades, and no alternation in the chemical structure was observed [3]. re-PVB was expected to contain both 3GO and DBS as plasticizers because of the reprocessing of various grades in the delamination process, but because of the overlapping with the characteristic peaks of the VB segment, it was not feasible to identify both [3].

Turning to TGA analysis (Figure 2a, Table 2), unplasticized PVB exhibited a single-step mass loss curve, corresponding to the decomposition of PVB. On the contrary, p-PVB and re-PVB presented two different mass loss steps, with the first referring to the evaporation of the contained plasticizers. The slightly high residue of re-PVB (0.4% *w*/*w* higher than p-PVB) could indicate the presence of a small fraction of glass residue in comparison with p-PVB.

Glass transition temperature (*T*_g_) values of each polymer were evaluated by typical DSC analysis under a nitrogen atmosphere. Unplasticized PVB exhibited a *T*_g_ value of ca. 71 °C, while plasticized samples exhibited significantly decreased *T*_g_ in the range of ca. 15–16 °C due to the presence of plasticizer (Table 3). Regarding the determination of MFR (Table 3), p-PVB and re-PVB presented similar values, in line with their plasticizer content as determined by TGA (Table 2), while u-PVB, which contained no plasticizer, exhibited a much lower value and required increased weight (10 kg instead of 2.16 kg) to flow.

### 3.2. Determination of PVB Degradation Mechanism

For the determination of the degradation mechanism of PVB, successive thermal (high temperature) and mechanical stresses (shearing) under the melt state in the internal mixer were applied to both p-PVB and u-PVB grades. The different PVB grades were selected to assess any possible interaction (positive or negative) of the plasticizer on the degradation mechanism. The received products, after kneading in the internal mixer for both p-PVB and u-PVB at 5, 10, and 15 min residence time, were evaluated through MFR, DSV, and YI. Figure 3a,b illustrate the effect of the increasing residence time on melt flow index and intrinsic viscosity, respectively. As shown in Figure 3a, in both PVB grades, MFR increased with processing time, proving that both grades suffered from thermal degradation. For u-PVB the MFR increase reached a ca. sixfold increase after 15 min of processing, while for p-PVB, the corresponding increase was limited to a ca. threefold value. The mechanical stresses were higher as melt viscosity increased, explaining why u-PVB showed a higher MFR increase. In fact, for the same quantity of p-PVB, the contained plasticizer fraction decreased the melt viscosity, resulting in reduced shear stresses in the system, thus limiting the overall suffered degradation. Similarly, as shown in Figure 3b, the IV values of both grades reduced with increasing processing time, an indirect sign of molecular weight decrease. The decrease in [*η*] was higher for u-PVB, as it suffered from higher mechanical stress. The combination of increasing MFR and meanwhile decreasing molecular weight (from IV) suggests a typical chain-scission degradation mechanism [24,43,57]. Both u-PVΒ and p-PVB showed the same trend in degradation, meaning that plasticizer might not chemically interfere in the thermo-oxidative degradation of PVB, but it can reduce the melt viscosity and/or absorb heat energy, therefore contributing to milder degradation [22,24]. According to macroscopic evidence, gradual yellowness was more distinct with increasing processing time for both PVB grades (Figure 3c), possibly because of the increased interaction with oxygen at a high temperature resulting in the formation of carbonyl species [22,58]. Once again, the total ΔYΙ was much higher for the case of u-PVB, again verifying that the pertinent grade suffered higher degradation under identical experimental conditions.

The thermally degraded products of u-PVB and p-PVB were also monitored by FT-IR analysis. Regarding u-PVB (Figure 4), no significant peak differentiations were observed, apart from the carbonyl region in the range of 1600–1800 cm^−1^. The peak therein corresponds to VAc, as explained in Figure 1, but a slight increase in its intensity with increasing processing time was observed, verifying the presence of new carbonyl species due to thermo-oxidative degradation. The spectrum change was evident but not as severe as that reported by El-Din and Sabaa [22]. Moreover, a new peak at ca. 1650 cm^−1^ appeared in the spectra of the 10- and 15-min processed samples, again assigned to degradation products (α,β-unsaturated ketones) as suggested by Grachev et al. [59]. This was not the case in the corresponding p-PVB samples (Appendix A), since on the one hand, the strong plasticizer peak overlapped at the same wavenumber range, while on the other hand, the lower suffered degradation due to the presence of the plasticizer might not result in the observation of new peaks.

### 3.3. Incorporation of AOs in PVB

Stabilizers are generally used at the end product of PVB with a common combination of primary and secondary antioxidants at a concentration ranging from 0.1 to 4.5% [28,30,60,61]. The retrieved PVB samples in the recycling process are expected to have a low residual content of antioxidants to protect the polymer during the reprocessing stage and during its next service life, because AOs can also be depleted during the service life of the material. Consequently, an appropriate formulation for the restabilization of PVB samples should be investigated. By utilizing the internal mixer, it is possible to experiment with small batches of PVB, enabling multiple screening experiments for the case of antioxidants. Six different AO formulations at a concentration of 0.3% *w/w* were tested for p-PVB and u-PVB at the residence time of 10 min, since while a ca. 5 min residence time is considered as a representative time similar to the industrial film processing time, the AOs should perform and inhibit thermo-oxidative degradation beyond the typical processing time, ensuring that they will not be depleted during the process but be available for the second life cycle of the material [16,24]. A 15 min residence time was considered an extreme condition to exhaust the efficiency of the tested AOs. The various formulations were tested in terms of MFR, OOT, and YI after the end of kneading.

For the case of u-PVB (Figure 5a), solely phenolic AOs (AO-1, AO-2, AO-3) inhibited the degradation to some extent in terms of MFR if compared with the unstabilized kneaded grade, but significant increases were still observed. On the contrary, when adding a secondary AO comprising thioether moieties to the already existing phenolic AO (AO-4) in a 1:1 ratio, the formulation performed exceptionally well, retaining MFR at the initial value. Similarly, when using a multifunctional additive comprising both phenolic and thioester moieties (AO-5), the degradation was again more successfully inhibited compared with the solely phenolic AO cases. This shows that thioester moieties contribute to stabilization, since apart from the peroxy radicals that are quenched by the hindered phenols, the formed hydroperoxides are decomposed by the complementary part of the antioxidant, thus substituting the need of adding extra secondary AOs (e.g., phosphite, thioester) [62]. AO-6 exhibited an intense decrease in MFR to the extent of not observing any melt flow during the measurement.

Furthermore, oxidation onset temperature (OOT) as determined by DSC analysis was used to evaluate the performance of antioxidants against oxidation. The usefulness of OOT as a tool for evaluating degradation is well documented, especially in the case of polyolefins, such as polypropylene [46,51,52,63]. Mei et al. also studied the thermal oxidative stability of three different primary antioxidants for PVB using a thermogravimetry–differential scanning calorimetry method and kinetic analysis [61]. As indicated by Figure 5a, the chain-scission mechanism caused a decrease in the oxidative thermal stability, causing a reduction in OOT from 219 °C in the neat u-PVB sample to 206 °C in the reprocessed grade without any AO added. Apart from AO-6, which exhibited the lowest thermal oxidative stability herein to the polymer (i.e., 202 °C), all the other AOs performed well, following the trend from MFR measurements (Appendix A).

In terms of yellowness, antioxidants did not provide sufficient color stability, and possibly because of alternations in the degradation mechanism, other degradation products could contribute to yellowness (Figure 5b). AO-5, despite its valuable MFR performance, presented inferior performance in terms of color efficiency. Thioether-containing antioxidants may induce yellowness or odor problems due to their sulfur content and their susceptibility to oxidation, but they are very effective in rubber-based plastics where color is not an issue [33]. All in all, AO-4 (containing 0.15% primary and 0.15% secondary AO) was the best antioxidant for u-PVB, since it showed the best MFR performance, retaining the value to its pristine amount and providing the highest oxidative stabilization to the system (Figure 5a), and maintained YI to acceptable levels.

Turning to the case of p-PVB, the antioxidants did not follow the same trend as in u-PVB. It can be shown (Figure 6a) that formulations with solely hindered phenols (primary AOs: AO-1, AO-2, AO-3) efficiently inhibited the degradation that the kneaded unstabilized sample suffered in terms of MFR, with AO-3 being the best performing among them. On the other hand, the grades that combined the presence of primary phenolic with secondary thioester (AO-4, AO-5, AO-6) exhibited similar stabilization performance in terms of MFR, with AO-6 showing the lowest value. However, AO-6 was excluded from the forthcoming analysis because of its poor OOT value (Figure 6a). Meanwhile, OOT showed AO-3, AO-4, and AO-5 (Figure 6a) as the best thermal oxidative stabilizers for p-PVB (Appendix A).

In literature, many phenolic antioxidants have been associated with the disadvantage of causing yellowing, depending on the processing conditions, which can be attributed to the reaction products of the antioxidants. The most commercially significant group of phenolic antioxidants for the production of PVB film contains the partial structure (3,5-di-tert-butyl-4-hydroxyphenyl)propionate, which is considered to lead to yellowing in PVB film with UV and/or temperature influence in the glass laminate [64]. Antioxidants containing this chemical structure, AO-1 and AO-2, were excluded for this reason, but also for their mediocre MFR performance. Correspondingly, AO-4, despite its good performance in terms of MFR (only a 10% increase) and OOT, displayed haziness that was observed macroscopically. Apart from yellowness, haze is also crucial, because it can significantly affect transparency, which is a critical parameter for the interlayer application. The presence of haziness indicates solubility issues between plasticizer and additives. In the literature, it has been shown that the solubility of the antioxidant in PVB is not a linear function of the plasticizer concentration in the polymer [65]. The additives can be soluble at high temperatures in the polymer melt but insoluble at low temperatures in the polymer film, thus giving rise to the formation of a separate phase within the film. Turbidity of PVB containing antioxidants at low temperatures has been documented with phenolic and phosphorous antioxidants in plasticized PVB [65]. In this particular case, haziness was attributed to the presence of the secondary sulfur-containing antioxidant. In terms of combined MFR, OOT, and YI performance, AO-3 and AO-5 were considered the best-performing ones for reprocessing PVB. Overall, the performance of each AO strongly depends on the PVB system (plasticized or unplasticized), but only plasticized PVB is associated with the recycling of PVB interlayer material.

Consequently, the best-performing AO formulations for p-PVB were also reevaluated for their applicability to re-PVB, a real PVB waste. Accordingly, three (3) AO formulations (AO-3, AO-4, and AO-5) at an incorporation level of 0.3 wt% were applied to re-PVB, and their effectiveness was gauged in terms of MFR and YI for 10 min residence time in the internal mixer at 150 °C and 40 rpm screw speed. MFR results (Figure 7a) supported the occurrence of a chain-scission mechanism for the neat re-PVB compared with the kneaded one (36% increase of MFR), proving that the residual additives were indeed insufficient to protect the polymer [40]. AO-5 exhibited the best performance in retaining the MFR value, while AO-3 also showed significant inhibition of degradation.

The yellowness index was not improved with the addition of additives (Figure 7b), which might have been due to the degradation products of AOs or their interactions with PVB and plasticizer during degradation. Nevertheless, YI remains an important factor, since re-PVB is intended for closed-loop recycling; therefore, high transparency and a clear view are strongly required. re-PVB with AO-3 was the best-performing additive formulation, and AO-5 came next, exactly as in the case of p-PVB. Especially for the case of AO-4, it was observed macroscopically that haze/turbidity was again introduced in the system despite the different composition of re-PVB (various high-quality PVB grades and different plasticizers). This case indicated again the immiscibility of AO-4 with PVB and plasticizer, rendering it inappropriate for further exploitation in interlayer applications.

From this study, it was concluded that the most effective AOs were AO-3 and AO-5 for both plasticized polymers (p-PVB and re-PVB), and this result points out that they had the same effect on plasticized PVB independently of their pristine PVB grade (p-PVB or re-PVB) or plasticizer type/content. Antioxidants exhibit universal applicability (Appendix A) in a plasticized PVB system, and their effectiveness can be evident when there is a prior categorization system (content of VB or plasticizer content).

### 3.4. Determination of Optimum AO Concentration in p-PVB

It is essential to optimize the consumption of AO in combination with the maximum performance to achieve a cost-effective recycling strategy while avoiding any compatibility issues that could affect the optical properties of the film. Therefore, two new AO concentrations (0.1% and 0.5% *w*/*w*) were tested only for p-PVB in three different residence times (5, 10, and 15 min) at 150 °C and 40 rpm in the internal mixer and compared with the results already shown for 0.3% concentration. The best-performing AOs from the previous section (AO-3 and AO-5) were selected for the optimization tests, and the results were gauged in terms of MFR, intrinsic viscosity (IV), and yellowness index (YI). Figure 8 illustrates the effectiveness of AO-3, indicating that a low concentration of AO (0.1%) was considered unfavorable, since the MFR and [*η*] at all residence times did not reach the required level. On the other hand, 0.5% *w*/*w* did not provide any significant improvement in MFR or [*η*]; therefore, it is considered superfluous. Turning to AO-5, the results of the study for 0.1% *w*/*w* were considered encouraging (Figure 9a,b), because a general steady performance similar to that of 0.3% was observed in the various residence times. For higher concentrations (0.5% *w*/*w*) of AO-5, a slightly improved performance was observed, but it did not compensate for the higher quantity. Overall, for both antioxidants, increased concentration did not exhibit significant assistance to the retention of MFR or [*η*] in all residence times; thus, it was considered redundant.

Considering also the YI values (Figure 8c and Figure 9c) of neat and restabilized p-PVB, it can be observed that AO-3 could overall better inhibit the yellowness than AO-5, which, on the contrary, offered yellowness in the samples at most residence times and concentrations. This could have been due to its sulfur-containing chemical structure, which is not usually intended for applications where transparency or yellowness is crucial. In the case of 10 min of residence time, 0.3% of AO-3 and AO-5 both seemed to perform effectively in limiting yellowness. Lowering the concentration (0.1% *w*/*w*) of AO-3 or AO-5 in 10 min resulted in both cases in more yellowness (Figure 8c and Figure 9c), since the remaining AO concentration was insufficient to inhibit oxidation, as it was in MFR. Focusing on 5 min of residence time, which was more indicative of the reprocessing part, AO-3 performed similarly in all concentrations. However, 0.3% of AO-5 seemed to cause yellowness issues; therefore, a lower concentration for AO-5 is suggested, since in parallel, the performance in MFR remained constant even with reduced concentration. For that reason, it can be concluded that AO-5 offers higher performance in terms of MFR than AO-3, but AO-3 exhibits slightly better performance in terms of YI. In conclusion, by examining MFR, [*η*], and YΙ, 0.3% of AO-3 performed well, while a fine-tuning of 0.15% AO-5 is suggested, which would lower the cost, limit yellowness, and provide the required performance in MFR. It is worthwhile mentioning that the increased yellowness by AO-5 could be exploited for colored laminated glass applications (privacy laminated glass), and its disadvantage in yellowness might not be always considered unfavorable [66].

Additionally, the thermal oxidative stability of AO-3 and AO-5 was tested in the AO concentration study at the critical condition only (10 min, 150 °C and 40 rpm), proving that AO-5 not only performed better in MFR but provided better thermo-oxidative stability than AO-3 at all concentrations and residence times (Figure 10).

### 3.5. Upscaling Restabilization of Re-PVB Wastes

The upscaling of the process was implemented by mixing a virgin plasticized commercial PVB grade (Trosifol Clear) with high-quality re-PVB grade retrieved through mechanochemical treatment at a 60:40 ratio, in order to compensate for the increased yellowness observed when using 100% re-PVB and AO-3 or AO-5. Four different formulations were tested, as presented in Table 4. A reference sample (no AOs) and three different samples were compounded with the best performing AO at 0.3% AO-3 and AO-5, as well as a reduced AO-3 concentration of 0.15%, and extruded via a flat die into a roll. It is worth mentioning that Tests 1B and 2 were prepared with the same re-PVB batch, but a different re-PVB batch was used for Tests 3A and 3B. The plasticizer content of the prepared re-PVB samples was determined by TGA (Table 4) and was found in the range of 27.7–30.3 wt%. To test the efficiency of the additives in the industrial level of the process, 50 g of each sample was subjected to thermo-oxidative degradation in the internal mixer at 150 °C and 40 rpm for 15 min to exhaust the material, and the results were evaluated in terms of MFR and YI.

After the induced degradation, the restabilized samples demonstrated great performance in terms of MFR (Figure 11a) in comparison with the unstabilized sample, even if reducing the AO concentration to 0.15%. More specifically, Test 2 showed a low increase (17%) in MFR, and Test 3B, only an insignificant increase (2.5%) in MFR, while the unstabilized sample (Test 1B) presented a 77.5% increase. Again AO-5 exhibited better performance in retaining MFR than AO-3. Regarding YI (Figure 11b), restabilized samples minimized the effect of thermo-oxidative degradation, with AO-3 having better performance (ΔYI = 2.28%) than AO-5 (ΔYΙ = 5.02%), while the unstabilized sample had the worst performance (ΔYΙ = 5.71%). From the latter, it can be concluded that in the unstabilized sample (Test 1B), the already contained AOs from manufacture were depleted during processing of the film. On the contrary, in the case of the stabilized samples, the extra AOs that were added were still active and performed well in inhibiting thermo-oxidative degradation during kneading in the internal mixer, implying that the AOs would remain effective during the second life cycle of the interlayer film.

Finally, the mechanical properties of the industrially produced films are shown in Table 5. It can be concluded that the mechanical properties of the films slightly varied when using different batches of re-PVB. Nevertheless, the values matched other commercial interlayer film specifications available in the market, which are found in the ranges of 23–25 MPa for tensile strength and 250–266% for elongation at break [67,68]. Furthermore, Test 3A and Test 3B (films produced with same re-PVB batch) showed that different antioxidants had similar, if not insignificant, effects on the mechanical properties of the films. Overall, it can be concluded that the addition of 40% of re-PVB along with the appropriate AOs for restabilization did not affect the interlayer production process or the quality of the final film and thus constituted a viable and sustainable solution for the closed-loop recycling of PVB.

## 4. Conclusions

In this study, a comprehensive approach to the recycling of polyvinyl butyral (PVB) wastes from laminated glasses after their end of life was developed. Thermo-oxidative degradation of PVB in the internal mixer at 150 °C, 40 rpm and different residence times (5, 10, 15 min) confirmed a chain-scission degradation mechanism for PVB by evaluation of MFR, [*η*], and YI. Six different antioxidant (AO) formulations were evaluated for their effectiveness in inhibiting the degradation that occurs at the critical condition of 150 °C and 40 rpm for 10 min. For u-PVB, AO-4 was the most effective but led to haziness in plasticized PVB, indicating an immiscibility issue of antioxidants with PVB and plasticizer. On the other hand, AO-3 and AO-5 were the best-performing AOs for p-PVB. AO-5 provided overall better restabilization concerning MFR and OOT, but AO-3 was more efficient as regards color stability. For the case of the retrieved PVB (re-PVB), AO-3 and AO-5 were determined again as the best-performing AOs, validating the universal applicability of the antioxidants for plasticized PVB systems. Tests for the optimum AO concentration suggested that 0.3% *w*/*w* was the most efficient for both AO-3 and AO-5, although acceptable performance could be achieved with 0.1% *w*/*w* of AO-5. For the pilot production of recycled PVB, virgin commercial plasticized PVB material was melt-mixed with re-PVB wastes in a 60/40 ratio with the addition of AO-3 or AO-5. The efficiency of the additives was also tested in that case in an exhausting condition (150 °C, 40 rpm, 15 min) in an internal mixer, and this proved once again that both AOs could successfully protect the polymer from degradation by retaining MFR and YI. Finally, the determined mechanical properties on the final interlayer films containing 40% recycled material were found in line with commercial specifications, thus rendering the pertinent strategy as a viable and sustainable solution for the closed-loop recycling of PVB.

## Figures and Tables

**Figure 1 polymers-17-00317-f001:**
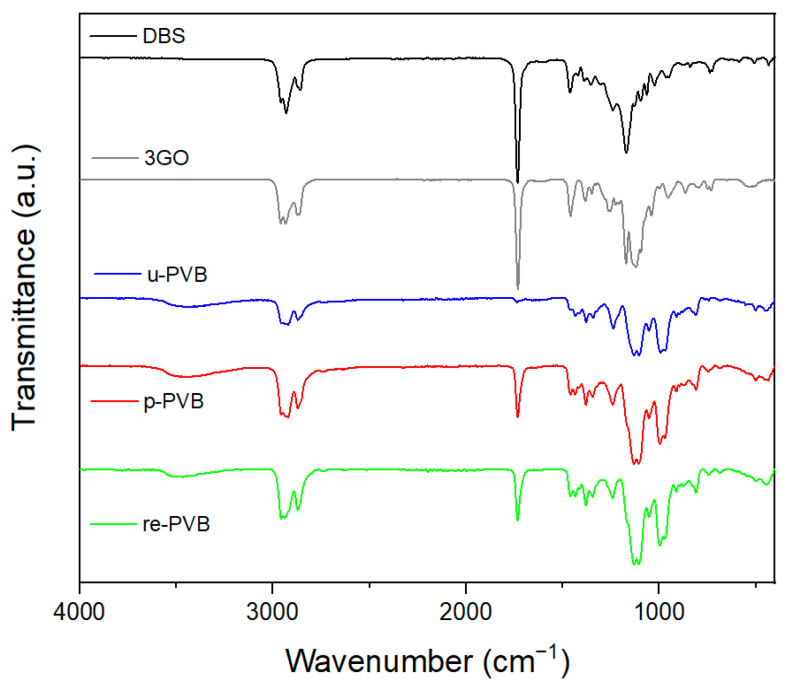
FT-IR spectra of the reference PVB materials (u-PVB, p-PVB, re-PVB) and the most common PVB plasticizers (DBS, 3GO).

**Figure 2 polymers-17-00317-f002:**
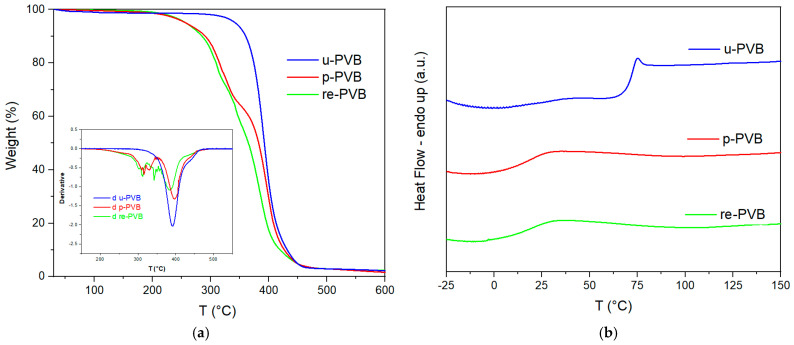
(**a**) Mass loss curves and derivative TGA curves of reference PVB grades; (**b**) 2nd heating DSC curves of reference PVB grades.

**Figure 3 polymers-17-00317-f003:**
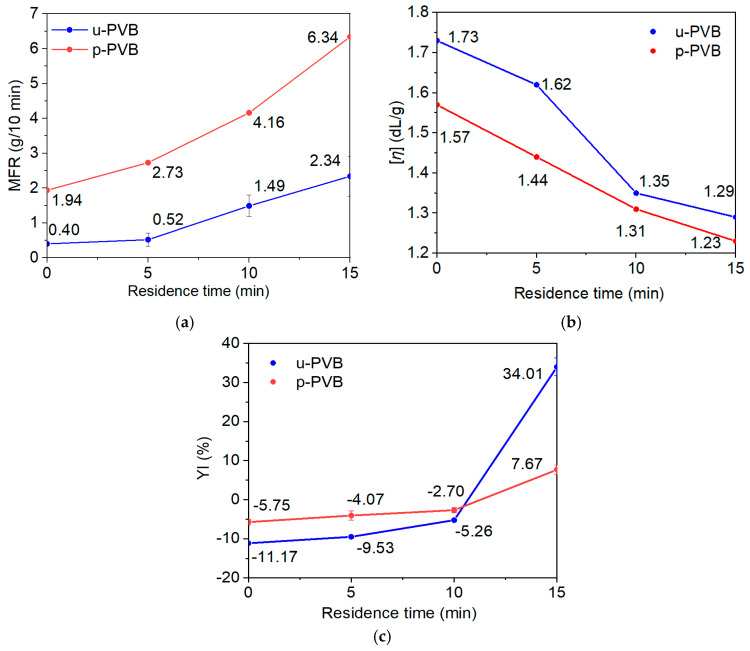
(**a**) Melt flow rate, (**b**) intrinsic viscosity [*η*], and (**c**) yellowness Index (YI) for u-PVB and p-PVB at different residence times.

**Figure 4 polymers-17-00317-f004:**
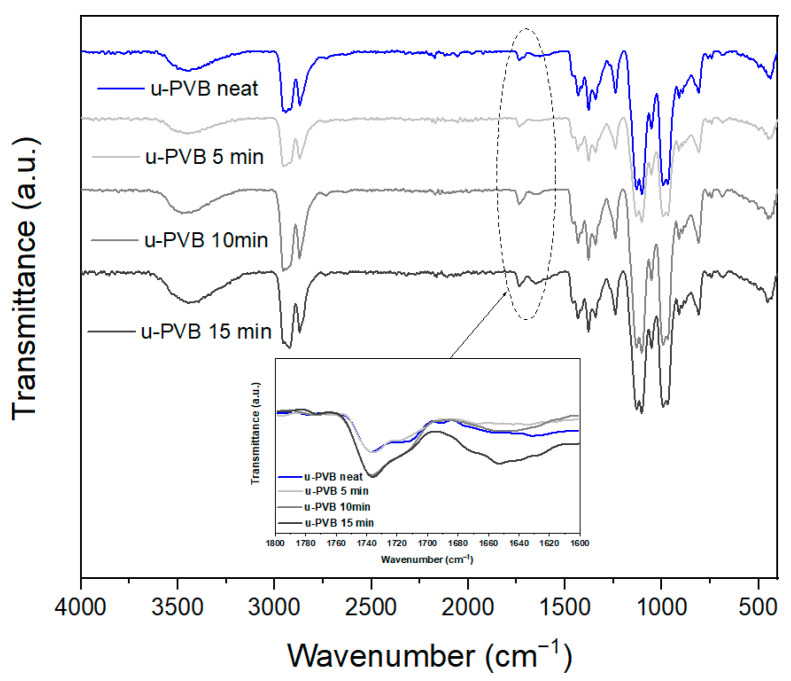
FT-IR spectra of u-PVB samples before and after kneading in the internal mixer.

**Figure 5 polymers-17-00317-f005:**
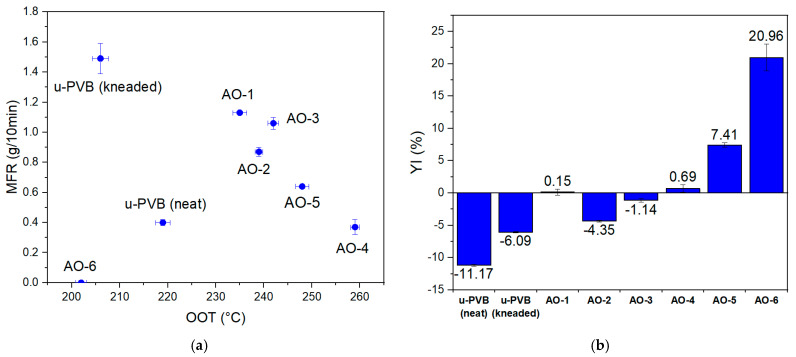
(**a**) MFR and OOT; (**b**) YI for the neat, unstabilized, and incorporated AOs (AO 1–6) for u-PVB in the critical condition of 10 min in the internal mixer.

**Figure 6 polymers-17-00317-f006:**
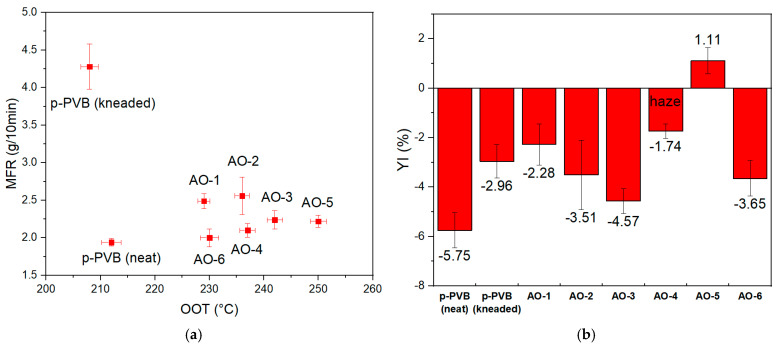
(**a**) MFR and OOT; (**b**) YI for the neat, unstabilized, and incorporated AOs (AO 1–6) for p-PVB in the critical condition of 10 min in the internal mixer.

**Figure 7 polymers-17-00317-f007:**
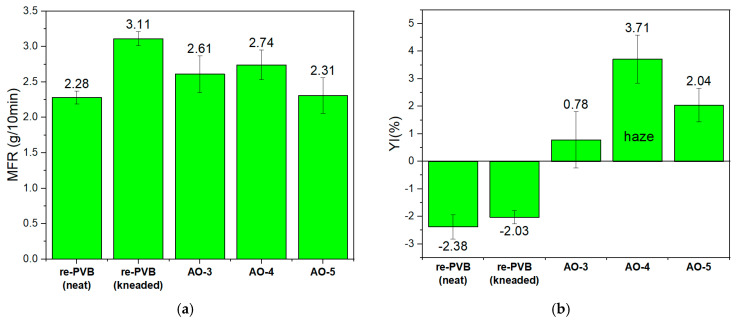
(**a**) MFR and (**b**) YI for the neat, unstabilized, and incorporated AO-3, AO-4, and AO-5 for re-PVB in the critical condition of 10 min, 40 rpm, 150 °C in the internal mixer.

**Figure 8 polymers-17-00317-f008:**
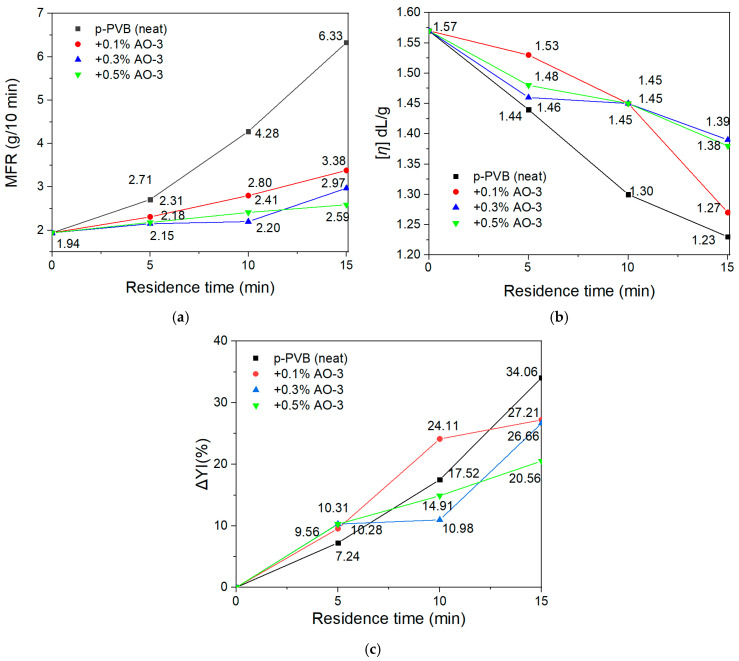
(**a**) MFR, (**b**) [*η*], and (**c**) YΙ for p-PVB with AO-3 at different concentrations (0.1%, 0.3%, 0.5% *w*/*w*) and residence times (0, 5, 10, 15) at 150 °C and 40 rpm in the internal mixer.

**Figure 9 polymers-17-00317-f009:**
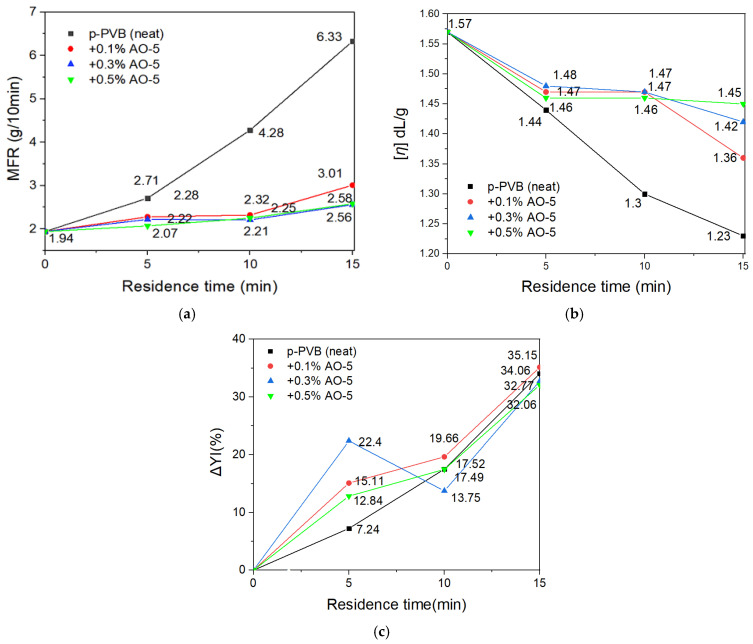
(**a**) MFR, (**b**) [*η*], and (**c**) YΙ for p-PVB with AO-5 at different concentrations (0.1%, 0.3%, 0.5% *w*/*w*) and residence times (0, 5, 10, 15) at 150 °C and 40 rpm in the internal mixer.

**Figure 10 polymers-17-00317-f010:**
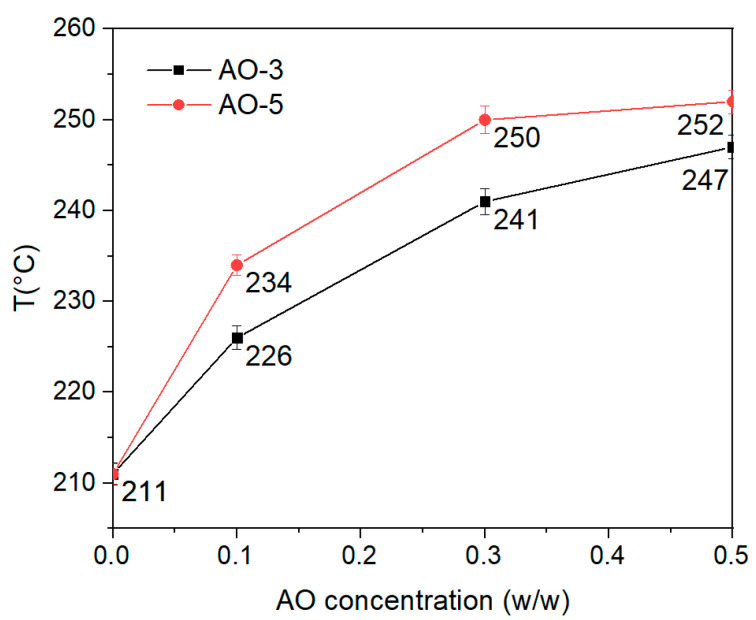
OOT of p-PVB with AO-3 and AO-5 for the different tested concentrations (0.1%, 0.3%, 0.5% *w*/*w*) at the critical condition (10 min, 150 °C and 40 rpm),.

**Figure 11 polymers-17-00317-f011:**
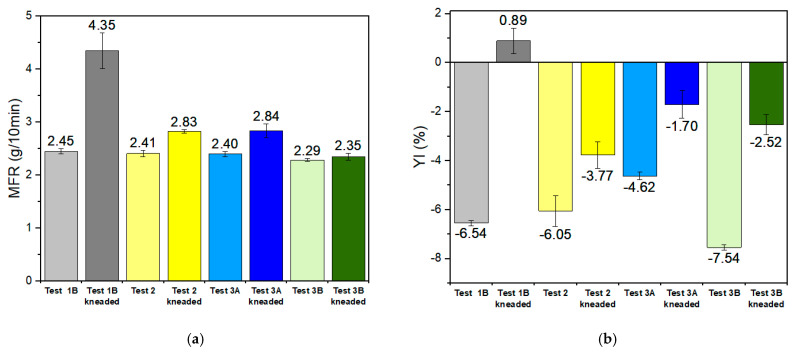
(**a**) MFR and (**b**) YI for Test 1B, Test 2, and Test 3B before and after being subjected to thermo-oxidative degradation in the internal mixer at 150 °C, 40 rpm for 15 min.

**Table 1 polymers-17-00317-t001:** Description of the incorporated antioxidants (AOs).

AO Formulation	Additives	Type
AO-1	octadecyl 3-(3,5-di-tert-butyl-4-hydroxyphenyl)propionate	Primary-hindered phenol
AO-2	pentaerythritol tetrakis(3-(3,5-di-tert-butyl-4-hydroxyphenyl)propionate)	
AO-3	1,3,5-trimethyl-2,4,6-tris(3,5-di-tert-butyl-4-hydroxybenzyl)benzene	
AO-4	AO-3 + octadecyl 3-[[3-(dodecyloxy)-3-oxopropyl]thio]propionate	1:1 combination of primary-hindered phenol and secondary thioester moieties
AO-5	4-[[4,6-bis(octylsulfanyl)-1,3,5-triazin-2-yl]amino]-2,6-di-tert-butylphenol	Multifunctional AO comprising phenol and thioether moieties
AO-6	2-methyl-4,6-bis[(octylthio)methyl]phenol	

**Table 2 polymers-17-00317-t002:** Results of TGA for u-PVB, p-PVB, and re-PVB.

Samples	*T*_5%_ (°C)	Step 1 (% *w*/*w*)	*T*_d1_ (°C)	*T*_d2_ (°C)	Step 2 (% *w*/*w*)	*T*_d3_ (°C)	Residue (% *w*/*w*)
u-PVB	329.1 ± 13.1	-	391.1 ± 0.4	-	99.0 ± 1.0	391.1 ± 0.4	1.0 ± 1.0
p-PVB	257.6 ± 0.5	35.8 ± 0.2	314.6 ± 1.8	-	60.2 ± 0.7	396.6 ± 1.0	2.2 ± 0.1
re-PVB	261.7 ± 4.5	39.4 ± 4.4	313.8 ± 1.9	341.3 ± 2.5	57.4 ± 4.1	388.7 ± 4.5	2.6 ± 0.1

**Table 3 polymers-17-00317-t003:** *T*_g_ and MFR values from DSC analysis for the reference samples.

Samples	*T*_g_ (°C)	MFR (g/10 min)
u-PVB	71.3 ± 1.3	0.40 ± 0.02 *
p-PVB	16.9 ± 0.7	1.94 ± 0.05 **
re-PVB	15.4 ± 1.1	2.28 ± 0.09 **

* measured with 10 kg load, ** measured with 2.16 kg load.

**Table 4 polymers-17-00317-t004:** Plasticizer content and specifications of the four different formulations of recycled PVB samples.

Sample	Plasticizer Content (%)	Specifications
Test 1B	29.7 ± 0.7	60% virgin commercial grade + 40% re-PVB (no AO)
Test 2	27.7 ± 1.2	60% virgin commercial grade + 40% re-PVB + 0.3% AO-3
Test 3A	28.0 ± 0.1	60% virgin commercial grade + 40% re-PVB + 0.15% AO-3
Test 3B	30.3 ± 0.2	60% virgin commercial grade + 40% re-PVB + 0.3% AO-5

**Table 5 polymers-17-00317-t005:** Mechanical properties of cast extruded films.

Sample	Tensile Strength at Break (MPa)	Elongation at Break (%)
Test 1B	30.6 ± 2.0	361.2 ± 17.5
Test 2	30.5 ± 7.1	276.8 ± 14.5
Test 3A	24.8 ± 1.4	345.0 ± 20.3
Test 3B	23.4 ± 0.7	336.5 ± 14.5

## Data Availability

Data are contained within the article and Appendix A.

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
