# Peer review of "Closed-Loop Recycling of Poly(vinyl butyral) Interlayer Film via Restabilization Technology"

_polymers, 2025, doi:10.3390/polym17030317_

Round 1

Reviewer 1 Report

Comments and Suggestions for Authors

The authors have reported remelting-re-stabilization of PVB using various antioxidants. Extensive experimental study various AOs have been reported. The authors have optimized the use of AOs for optimum performance.  The manuscript may be improved by addressing the following comments.

1.     How does AOs affect the TGA and DSC analysis?

2.     Will using this re-stabilized PVB be cost-effective compared to the virgin one?

3.     As the authors have mentioned the possible use of this polymer in the windscreen, is there any analysis to support the impact analysis of this restabilized PVB.

Author Response

Point 1.     How does AOs affect the TGA and DSC analysis?

Response:  Antioxidants, as also mentioned in the text in lines 136-149, are typical processing stabilizers that act as radical scavengers (primary) or as hydroperoxide decomposers (secondary) at high temperatures (e.g. during processing), thus inhibiting thermo-oxidation of the polymer. The latter will result in an increased thermal stability. Consequently, regarding TGA the incorporation of AOs would slightly increase the thermal stability of the polymer (onset decomposition temperature, Td5% and maximum decomposition temperature, Td). The effect of AOs can be also monitored by TGA via conducting kinetic analysis. The Kissinger method is a useful approach to evaluate the activation energy of thermal degradation, including processes influenced by antioxidants, using data from Thermogravimetric Analysis (TGA). Accordingly, the determined activation energy of the thermal decomposition of the AO-containing compounds is always higher compared to the un-stabilized ones. This is well documented in literature (Mei et. al, DOI: 10.1007/s10973-014-3806-3 and Ivanov et. al https://doi.org/10.1016/j.tca.2014.05.016), but that approach was not studied on this paper.

For the case of DSC, no alternation in the glass transition temperature (Tg) is expected for the low concentrations of AOs used herein (0.1-0.5 wt%), especially in the case of p-PVB samples, which already contain significant amounts of plasticizer. However, by using DSC to determine Oxidation Onset Temperature (OOT) in an oxidative environment (air flow), the effect of the various antioxidants was clear, exhibiting increased OOT values in the range of 17-40°C as showed in the manuscript (Figure 5a, Figure 6a) and in the supplementary material (Figure S3). The OOT increase is directly connected with the inhibition of oxidation and its shift to higher temperature, due to the presence of the AOs.

Point 2.     Will using this re-stabilized PVB be cost-effective compared to the virgin one?

Response:  The results of this publication were generated within SUNRISE project that has received funding from the European Union’s Horizon 2020 Research and Innovation Program under grant agreement No 958243. Within this project there have been various Life cycle assessments (LCA) conducted for re-stabilized PVB from respective partners on the field. These studies showed that PVB recycling remains environmentally more beneficial than PVB primary production regarding carbon footprint, human non-carcinogenic toxicity and resources consumption. More specifically, the data from Sunrise Project (https://sunrise-project.eu/) reveal that re-PVB flakes are priced at €900/ton, while antioxidants cost €17,000/ton with only 0.3% w/w being utilized. On the contrary, virgin PVB flakes can cost up to 10,000€/ton proving that there is a high profit margin for PVB recycling. Additionally, the use of 40% w/w recycled PVB in the end product is a relatively high recycled content. Therefore, a 40% substitution of primary PVB production can be indeed cost-effective and financially beneficial, as long as processing issues (as described herein) are mitigated. Meanwhile, laminated glass wastes (from automotive and/or architectural applications) can be fully exploited, since glass recyclers can fully recycle the glass, while also through this study a strategy towards closed loop recycling is shown, fully in line with the principles of circular economy. 

Point 3.     As the authors have mentioned the possible use of this polymer in the windscreen, is there any analysis to support the impact analysis of this restabilized PVB.

Response: Pummel tests are typically used for assessing the impact resistance of PVB after its lamination with glass. Within the holistic scope of this project, laminated PVB samples were produced and tested via Pummel test for re-PVB impact resistance efficiency. The restabilized samples (Test 2, Test 3A, Test 3B), mentioned in the reviewed paper, ranged from moderate Pummel values (3-5) even to high values (6-8), proving their sufficient adhesion to glass and fulfilling the safety requirements (Table 1). The recommended pummel range for conventional interlayers and laminated glass is 4 – 7 as determined on glass ≤ 4mm thick (suggested by Eastman Company).

Table 1: Tested samples and their value in the Pummel test.

Samples

Pummel Value

Test 1B

7

Test 2

5-7

Test 3A

Not homogeneous, 3-5 in different areas

Test 3B

6

Nevertheless, this paper focuses only on the reprocessing and re-stabilization of the retrieved PVB flakes and the impact performance of the laminated glass is not part of this study. It is worth mentioning that Pummel test or any other impact resistance test on the laminated glass is also affected by the lamination process, which can be modified to achieve higher or lower adhesion in certain cases. In addition, other special additives, i.e. adhesion controllers, can be incorporated to PVB in order to achieve the desired pummel value.

Reviewer 2 Report

Comments and Suggestions for Authors

This article deals with an interesting topic related to the analysis of the recycling process of post-consumer polyvinyl butyral waste. The manuscript is written in a clear style, but some points need clarification.

·         It is advisable not to use abbreviations in the title of the article.

·         For scientific accuracy and to understand the behaviour of PVB, it is necessary to provide the amount of plasticiser in plasticized and recycled PVB compositions.

·         The caption of Figure 1 is unclear and should be corrected.

·         Lines 316, 323: ‘Figure 2’ should be replaced by ‘Figure 3’.

·         It is not clear what the origin of virgin commercial PVB (Section 4.5). Is it commercial grade plasticized or unplasticized PVB?

·         The caption of Table 4 is incorrect.

·         How are PVB samples for mechanical testing? What are the parameters of the mechanical test?

·         In my opinion, it would be more accurate if the mechanical properties of the virgin commercial grade PVB film were determined rather than taken from the technical data sheet (Table 5).

·         The conclusions are too long and confusing. They should be rewritten.

Author Response

Point 1. It is advisable not to use abbreviations in the title of the article.

Response: Page 1, Line 2: Title changed  to ‘Closed-loop Recycling of poly(vinyl butyral) Interlayer Film via Re-stabilization Technology

Point 2. For scientific accuracy and to understand the behaviour of PVB, it is necessary to provide the amount of plasticiser in plasticized and recycled PVB compositions.

Response: Plasticizer content (1st mass loss step in the TGA mass loss curves, Figure 2a, page 8 top) of p-PVB & re-PVB was already determined by TGA and was also mentioned in Table 2 (Page 7, bottom). Plasticizer content of the upscaled recycled PVB samples (Test 1B, Test 2, Test 3A & Test 3B) was also determined by TGA analysis and the respective values were added in Table 4 (Page 16 bottom, Line 538) and a small comment was added in lines 532-533 in the text.

Point 3. The caption of Figure 1 is unclear and should be corrected.

Response: The caption of Figure 1 was updated (Page 7, Line 288-289) to ‘Figure 1: FT-IR spectra of the reference PVB materials (u-PVB, p-PVB, re-PVB) and the most common PVB plasticizers (DBS, 3GO)’.  

Point 4. Lines 316, 323: ‘Figure 2’ should be replaced by ‘Figure 3’.

Response: Corrected appropriately at (now) Line 317 & Line 324

Point 5. It is not clear what the origin of virgin commercial PVB (Section 4.5). Is it commercial grade plasticized or unplasticized PVB?

Response: The origin of virgin commercial PVB was already defined as .. virgin commercial PVB grade (Trosifol Clear, Kuraray)..’ in Page 6, Line 243. The word ‘plasticized’ was also added in that sentence for clarity purposes. Additionally, the sentence in Page 16, Lines 525-526 was changed accordingly to ‘The upscaling of the process was implemented by mixing a virgin plasticized commercial PVB grade (Trosifol Clear) with high quality re-PVB grade… ‘ to be more clear for the reader.

Point 6. The caption of Table 4 is incorrect.

Response: Updated Table caption (Page 16, Line 539) to ‘Table 4: Plasticizer content and specifications of the four different formulations of recycled PVB samples’. In addition, the plasticizer content of recycled PVB samples was added in Table 4.

Point 7. How are PVB samples for mechanical testing? What are the parameters of the mechanical test?

Response:  The dimensions of PVB samples and the parameters of mechanical testing for the pilot scale samples are described in the paper from Line 256 to Line 262: The film samples were cut into rectangular stripes of 250x15 mm2 dimensions, the initial distance between the grips was 50 mm and a grip separation rate of 500 mm/min were used to obtain tensile strength at break and elongation at break. 9 samples of each grade were tested, and the results were represented as their average. All samples were pre-conditioned at 22 °C and 50% RH for 40h prior to measurement.’

Some information was added in the text and was highlighted in yellow so as to be more clear for the reader (lines 258, 261, 262).

Additional information is also provided here:

  • Method of film preparation: pneumatic press with a die cut mould.
  • All tests were done in the machine direction.
  • Tests were performed in the same room.

Point 8. In my opinion, it would be more accurate if the mechanical properties of the virgin commercial grade PVB film were determined rather than taken from the technical data sheet (Table 5).

Response: The properties of various plasticized commercial PVB grades were extracted from technical data sheets to emphasize their relation to the mechanical properties of recycled PVB discussed in this paper. The measurement of virgin commercial grade (Trosifol Clear) might not be so accurate because it is commercially available only in the form of flakes, therefore a compounding step was needed prior to its film production and the respective yield of specimens for mechanical testing. This reprocessing stage (without AOs) might result in limited degradation to the final PVB film grade and thus its value might not be accurate. However, the mechanical properties of this grade were measured after reprocessing and listed below in Table 1.

Table 1: Mechanical properties of commercial plasticized PVB grade (Trosifol Clear) and the other recycled PVB grades mentioned in the paper

Samples

Tensile Strength at Break (MPa):

Elongation at Break (%):

Trosifol Clear

22.9 ± 1.2 MPa

363.4 ± 28.1 %

Test 1B

30.6 ± 2.0

361.2 ± 17.5

Test 2

30.5 ± 7.1

276.8 ± 14.5

Test 3A

24.8 ± 1.4

345.0 ± 20.3

Test 3B

23.4 ± 0.7

336.5 ± 14.5

The values of the reprocessed Trosifol Clear flakes are again in the range of 23-25 MPa for tensile strength and above 250-266 % for elongation at break as mentioned in the paper and in the technical datasheets found online. However, the values of this PVB grade were omitted from the text because it lacks the same reprocessing as the other recycled PVB grades.

Point 9. The conclusions are too long and confusing. They should be rewritten.

Response: The conclusions were shortened and they were kept at a more reasonable length. Changes on Page 18, from Lines 571 to 593.

Round 2

Reviewer 2 Report

Comments and Suggestions for Authors

Thanks to the authors for the changes to the manuscript. The overall quality has improved and, in my opinion, the manuscript can be accepted for publication in Polymers.